# Best Possible Q-Learning

## Abstract

Fully decentralized learning, where the global information, *i.e.*, the actions of other agents, is inaccessible, is a fundamental challenge in cooperative multi-agent reinforcement learning. However, the convergence and optimality of most decentralized algorithms are not theoretically guaranteed, since the transition probabilities are non-stationary as all agents are updating policies simultaneously. To tackle this challenge, we propose *best possible operator*, a novel decentralized operator, and prove that the policies of cooperative agents will converge to the optimal joint policy if each agent independently updates its individual state-action value by the operator when there is only one optimal joint policy. Further, to make the update more efficient and practical, we simplify the operator and prove that the convergence and optimality still hold with the simplified one. By instantiating the simplified operator, the derived fully decentralized algorithm, *best possible Q-learning* (BQL), does not suffer from non-stationarity. Empirically, we show that BQL achieves remarkable improvement over baselines in a variety of cooperative multi-agent tasks.

## 1 Introduction

Cooperative multi-agent reinforcement learning (MARL) trains a group of agents to maximize the cumulative shared reward, which has great significance for real-world applications, including logistics (Li et al., 2019), traffic signal control (Xu et al., 2021), power dispatch (Wang et al., 2021), and games (Vinyals et al., 2019). Although most existing methods follow the paradigm of centralized training and decentralized execution (CTDE), in many scenarios where the information of all agents is unavailable in the training period, each agent has to learn independently without centralized information. Thus, *fully decentralized learning*, where the agents can only use local experiences without the actions of other agents, is highly desirable (Jiang & Lu, 2022).

However, in fully decentralized learning, as other agents are treated as a part of the environment and are updating their policies simultaneously, the transition probabilities from the perspective of individual agents will be non-stationary. Thus, the convergence of most decentralized algorithms, *e.g.*, independent Q-learning (IQL) (Tan, 1993), is not theoretically guaranteed. Multi-agent alternate Q-learning (MA2QL) (Su et al., 2022) guarantees the convergence to a Nash equilibrium, but the converged equilibrium may not be the optimal one when there are multiple equilibria (Zhang et al., 2021a). Distributed IQL (Lauer & Riedmiller, 2000) and I2Q (Jiang & Lu, 2022) can learn the optimal joint policy, yet are limited to deterministic environments. How to guarantee the convergence of the optimal joint policy in *stochastic environments* remains open.

To tackle this challenge, we propose **best possible operator**, a novel decentralized operator to update the individual state-action value of each agent, and prove that the policies of agents converge to the optimal joint policy under this operator when there is only one optimal joint policy. However, it is inefficient and thus impractical to perform best possible operator, because at each update it needs to compute the expected values of all possible transition probabilities and update the state-action value to be the maximal one. Therefore, we further propose *simplified best possible operator*. At each update, the simplified operator only computes the expected value of one of the possible transition probabilities and monotonically updates the state-action value. We prove that the policies of agents also converge to the optimal joint policy under the simplified operator. We respectively instantiate the simplified operator with Q-table for tabular cases and with neural networks for complex environments. In the Q-table instantiation, non-stationarity is instinctively avoided, and in the neural

network instantiation, non-stationarity in the replay buffer is no longer a drawback, but a necessary condition for convergence.

The proposed algorithm, ***best possible Q-learning*** (**BQL**), is fully decentralized, without using the information of other agents. We evaluate BQL on a variety of multi-agent cooperative tasks, *i.e.*, stochastic games, MPE-based differential games (Lowe et al., 2017), Multi-Agent MuJoCo (de Witt et al., 2020b), SMAC (Samvelyan et al., 2019), and GRF (Kurach et al., 2020), covering fully and partially observable, deterministic and stochastic, discrete and continuous environments. Empirically, BQL substantially outperforms baselines. To the best of our knowledge, BQL is the first decentralized algorithm that guarantees the convergence to the global optimum in stochastic environments. More simplifications and instantiations of *best possible operator* can be further explored. We believe BQL can be a new paradigm for fully decentralized learning.

## 2 METHOD

### 2.1 PRELIMINARIES

Consider $N$-agent MDP(Oliehoek et al., 2016) $M_{\text{env}} = < \mathcal{S}, \mathcal{O}, \mathcal{A}, R, P_{\text{env}}, \gamma >$ with the state space $\mathcal{S}$ and the joint action space $\mathcal{A}$. Each agent $i$ chooses an individual action $a_i$, and the environment transitions to the next state $s'$ by taking the joint action $\boldsymbol{a}$ with the transition probabilities $P_{\text{env}}(s'|s, \boldsymbol{a})$. Although in theoretical analysis, we assume all agents obtain the state $s$, in practice each agent $i$ can make decisions using local observation $o_i \in \mathcal{O}$ or trajectory. All agents obtain a shared reward $r = R(s, s') \in [r_{\min}, r_{\max}]$ and learn to maximize the expected discounted return $\mathbb{E} \sum_{t=0}^{\infty} \gamma^t r_t$. In fully decentralized setting, $M_{\text{env}}$ is partially observable, since each agent $i$ only observes its own action $a_i$ instead of the joint action $\boldsymbol{a}$. From the perspective of each agent $i$, there is an MDP $M_i = < \mathcal{S}, \mathcal{A}_i, R, P_i, \gamma >$ with the individual action space $\mathcal{A}_i$ and the transition probabilities

$$P_i(s'|s, a_i) = \sum_{\boldsymbol{a}_{-i}} P_{\text{env}}(s'|s, a_i, \boldsymbol{a}_{-i}) \boldsymbol{\pi}_{-i}(\boldsymbol{a}_{-i}|s) \tag{1}$$

where $\boldsymbol{\pi}_{-i}$ denotes the joint policy of all agents except agent $i$, similarly for $\boldsymbol{a}_{-i}$. According to (1), the transition probabilities $P_i$ depend on the policies of other agents $\boldsymbol{\pi}_{-i}$. As other agents are updating their policies continuously, $P_i$ becomes *non-stationary*. On the non-stationary transition probabilities, the convergence of independent Q-learning[1]

$$Q_i(s, a_i) = \mathbb{E}_{P_i(s'|s, a_i)} \left[ r + \gamma \max_{a_i'} Q_i(s', a_i') \right] \tag{2}$$

is not guaranteed, and how to learn the optimal joint policy in fully decentralized settings is quite a challenge. In the next section, we propose *best possible operator*, a novel fully decentralized operator, which guarantees the convergence to the optimal joint policy in stochastic environments.

### 2.2 BEST POSSIBLE OPERATOR

First, let us consider the optimal joint Q-value

$$Q(s, \boldsymbol{a}) = \mathbb{E}_{P_{\text{env}}(s'|s, \boldsymbol{a})} \left[ r + \gamma \max_{\boldsymbol{a}'} Q(s', \boldsymbol{a}') \right], \tag{3}$$

which is the expected return of the optimal joint policy $\boldsymbol{\pi}^*(s) = \arg \max_{\boldsymbol{a}} Q(s, \boldsymbol{a})$. Based on the optimal joint Q-value, for each agent $i$, we define $\max_{\boldsymbol{a}_{-i}} Q(s, a_i, \boldsymbol{a}_{-i})$, which follows the fixed point equation:

$$\max_{\boldsymbol{a}_{-i}} Q(s, a_i, \boldsymbol{a}_{-i}) = \max_{\boldsymbol{a}_{-i}} \mathbb{E}_{P_{\text{env}}(s'|s, \boldsymbol{a})} \left[ r + \gamma \max_{a_i'} \max_{\boldsymbol{a}_{-i}'} Q(s, a_i', \boldsymbol{a}_{-i}') \right] \tag{4}$$

$$= \mathbb{E}_{P_{\text{env}}(s'|s, a_i, \boldsymbol{\pi}_{-i}^*(s, a_i))} \left[ r + \gamma \max_{a_i'} \max_{\boldsymbol{a}_{-i}'} Q(s, a_i', \boldsymbol{a}_{-i}') \right] \tag{5}$$

where $\boldsymbol{\pi}_{-i}^*(s, a_i) = \arg \max_{\boldsymbol{a}_{-i}} Q(s, a_i, \boldsymbol{a}_{-i})$ is the optimal conditional joint policy of other agents given $a_i$. (4) is from taking $\max_{\boldsymbol{a}_{-i}}$ on both sides of (3), and (5) is by folding $\boldsymbol{\pi}_{-i}^*(s, a_i)$ into $P_{\text{env}}$. Then we have the following lemma.

---

[1]For simplicity, we refer to the optimal value $Q^*$ as $Q$ in this paper, unless stated otherwise.

**Lemma 1.** *If each agent $i$ learns the independent value function $Q_i(s, a_i) = \max_{\boldsymbol{a}_{-i}} Q(s, a_i, \boldsymbol{a}_{-i})$, and takes actions as $\arg\max_{a_i} Q_i(s, a_i)$, the agents will obtain the optimal joint policy when there is only one optimal joint policy[2].*

*Proof.* As $\max_{a_i} \max_{\boldsymbol{a}_{-i}} Q(s, a_i, \boldsymbol{a}_{-i}) = \max_{\boldsymbol{a}} Q(s, \boldsymbol{a})$ and there is only one optimal joint policy, $\arg\max_{a_i} Q_i(s, a_i)$ is the action of agent $i$ in the optimal joint action $\boldsymbol{a}$. $\qquad\square$

According to Lemma 1, to obtain the optimal joint policy is to let each agent $i$ learn the value function $Q_i(s, a_i) = \max_{\boldsymbol{a}_{-i}} Q(s, a_i, \boldsymbol{a}_{-i})$. To this end, we propose *a new operator* to update $Q_i$ in a fully decentralized way:

$$Q_i(s, a_i) = \max_{P_i(\cdot|s,a_i)} \mathbb{E}_{P_i(s'|s,a_i)} \left[ r + \gamma \max_{a_i'} Q_i(s', a_i') \right]. \tag{6}$$

Given $s$ and $a_i$, there will be numerous $P_i(s'|s, a_i)$ due to different other agents' policies $\boldsymbol{\pi}_{-i}$. To reduce the complexity, we only consider the deterministic policies, because when there is only one optimal joint policy, the optimal joint policy must be deterministic (Puterman, 1994). So the operator (6) takes the maximum only over the transition probabilities $P_i(s'|s, a_i)$ under *deterministic $\boldsymbol{\pi}_{-i}$*. Intuitively, the operator continuously pursues the 'best possible expected return', until $Q_i$ reaches the optimal expected return $\max_{\boldsymbol{a}_{-i}} Q(s, a_i, \boldsymbol{a}_{-i})$, so we name the operator (6) **best possible operator**. In the following, we theoretically prove that $Q_i(s, a_i)$ converges to $\max_{\boldsymbol{a}_{-i}} Q(s, a_i, \boldsymbol{a}_{-i})$ under best possible operator, thus the agents learn the optimal joint policy. Let $Q_i^k(s, a_i)$ denote the value function in the update $k$ and $Q_i(s, a_i) := Q_i^\infty(s, a_i)$. Then, we have the following lemma.

**Lemma 2.** *If $Q_i^0$ is initialized to be the minimal return $\frac{r_{\min}}{1-\gamma}$, $\max_{\boldsymbol{a}_{-i}} Q(s, a_i, \boldsymbol{a}_{-i}) \geq Q_i^k(s, a_i), \forall s, a_i, \forall k$, under best possible operator.*

*Proof.* We prove the lemma by induction. First, as $Q_i^0$ is initialized to be the minimal return, $\max_{\boldsymbol{a}_{-i}} Q(s, a_i, \boldsymbol{a}_{-i}) \geq Q_i^0(s, a_i)$. Then, suppose $\max_{\boldsymbol{a}_{-i}} Q(s, a_i, \boldsymbol{a}_{-i}) \geq Q_i^{k-1}(s, a_i), \forall s, a_i$. By denoting $\arg\max_{P_i(s'|s,a_i)} \mathbb{E}_{P_i(s'|s,a_i)} \left[ r + \gamma \max_{a_i'} Q_i^{k-1}(s', a_i') \right]$ as $P_i^*(s'|s, a_i)$, we have

$$\max_{\boldsymbol{a}_{-i}} Q(s, a_i, \boldsymbol{a}_{-i}) - Q_i^k(s, a_i)$$

$$= \max_{\boldsymbol{a}_{-i}} \sum_{s'} P_{\text{env}} \left( s'|s, a_i, \boldsymbol{a}_{-i} \right) \left[ r + \gamma \max_{a_i'} \max_{\boldsymbol{a}_{-i}'} Q(s', a_i', \boldsymbol{a}_{-i}') \right] - \sum_{s'} P_i^*(s'|s, a_i) \left[ r + \gamma \max_{a_i'} Q_i^{k-1}(s', a_i') \right]$$

$$\geq \sum_{s'} P_i^*(s'|s, a_i) \left[ r + \gamma \max_{a_i'} \max_{\boldsymbol{a}_{-i}'} Q(s', a_i', \boldsymbol{a}_{-i}') \right] - \sum_{s'} P_i^*(s'|s, a_i) \left[ r + \gamma \max_{a_i'} Q_i^{k-1}(s', a_i') \right]$$

$$= \gamma \sum_{s'} P_i^*(s'|s, a_i) \left( \max_{a_i'} \max_{\boldsymbol{a}_{-i}'} Q(s', a_i', \boldsymbol{a}_{-i}') - \max_{a_i'} Q_i^{k-1}(s', a_i') \right)$$

$$\geq \gamma \sum_{s'} P_i^*(s'|s, a_i) \left( \max_{\boldsymbol{a}_{-i}'} Q(s', a_i'^*, \boldsymbol{a}_{-i}') - Q_i^{k-1}(s', a_i'^*) \right) \geq 0,$$

where $a_i'^* = \arg\max_{a_i'} Q_i^{k-1}(s', a_i')$. Thus, it holds in the update $k$. By the principle of induction, the lemma holds for all updates. $\qquad\square$

Intuitively, $\max_{\boldsymbol{a}_{-i}} Q(s, a_i, \boldsymbol{a}_{-i})$ is the optimal expected return after taking action $a_i$, so it is the upper bound of $Q_i(s, a_i)$. Further, based on Lemma 2, we have the following lemma.

**Lemma 3.** *$Q_i(s, a_i)$ converges to $\max_{\boldsymbol{a}_{-i}} Q(s, a_i, \boldsymbol{a}_{-i})$ under best possible operator.*

*Proof.* For clear presentation, we use $P_{\text{env}} \left( s'|s, a_i, \boldsymbol{\pi}_{-i}^* \right)$ to denote $P_{\text{env}} \left( s'|s, a_i, \boldsymbol{\pi}_{-i}^*(s, a_i) \right)$. From (5) and (6), we have

$$\left\| \max_{\boldsymbol{a}_{-i}} Q(s, a_i, \boldsymbol{a}_{-i}) - Q_i^k(s, a_i) \right\|_\infty = \max_{s,a_i} \left( \sum_{s'} P_{\text{env}} \left( s'|s, a_i, \boldsymbol{\pi}_{-i}^* \right) \left[ r + \gamma \max_{a_i'} \max_{\boldsymbol{a}_{-i}'} Q(s', a_i', \boldsymbol{a}_{-i}') \right] \right.$$

$$\left. - \sum_{s'} P_i^*(s'|s, a_i) \left[ r + \gamma \max_{a_i'} Q_i^{k-1}(s', a_i') \right] \right) \leftarrow (\text{Lemma 2})$$

---

[2]We can use the simple solution proposed in I2Q to deal with the limitation of only one joint policy, which is included in Appendix D.

$$\leq \max_{s,a_i} \left( \sum_{s'} P_{\text{env}} \left( s'|s,a_i, \boldsymbol{\pi}^*_{-i} \right) \left[ r + \gamma \max_{a_i'} \max_{\boldsymbol{a}'_{-i}} Q(s',a_i',\boldsymbol{a}'_{-i}) \right] \right.$$

$$\left. - \sum_{s'} P_{\text{env}} \left( s'|s,a_i, \boldsymbol{\pi}^*_{-i} \right) \left[ r + \gamma \max_{a_i'} Q_i^{k-1}(s',a_i') \right] \right)$$

$$\leq \gamma \max_{s',a_i'} \left( \max_{\boldsymbol{a}'_{-i}} Q(s',a_i',\boldsymbol{a}'_{-i}) - Q_i^{k-1}(s',a_i') \right)$$

$$= \gamma \left\| \max_{\boldsymbol{a}_{-i}} Q(s,a_i,\boldsymbol{a}_{-i}) - Q_i^{k-1}(s,a_i) \right\|_\infty.$$

We have $\left\| \max_{\boldsymbol{a}_{-i}} Q(s,a_i,\boldsymbol{a}_{-i}) - Q_i^k(s,a_i) \right\|_\infty \leq \gamma^k \left\| \max_{\boldsymbol{a}_{-i}} Q(s,a_i,\boldsymbol{a}_{-i}) - Q_i^0(s,a_i) \right\|_\infty$. Let $k \to \infty$, then $Q_i(s,a_i) \to \max_{\boldsymbol{a}_{-i}} Q(s,a_i,\boldsymbol{a}_{-i})$, thus the lemma holds. $\square$

According to Lemma 1 and 3, we immediately have:

**Theorem 1.** *The agents learn the optimal joint policy under best possible operator when there is only one optimal joint policy.*

## 2.3 SIMPLIFIED BEST POSSIBLE OPERATOR

Best possible operator guarantees the convergence to the optimal joint policy. However, to perform (6), every update, each agent $i$ has to compute the expected values of all possible transition probabilities and update $Q_i$ to be the maximal expected value, which is too costly. Therefore, we introduce an auxiliary value function $Q_i^{\text{e}}(s,a_i)$, and simplify (6) into two operators. First, at each update, we randomly select one of possible transition probabilities $\tilde{P}_i$ for each $(s,a_i)$ and update $Q_i^{\text{e}}(s,a_i)$ by

$$Q_i^{\text{e}}(s,a_i) = \mathbb{E}_{\tilde{P}_i(s'|s,a_i)} \left[ r + \gamma \max_{a_i'} Q_i(s',a_i') \right]. \tag{7}$$

$Q_i^{\text{e}}(s,a_i)$ represents the expected value of the selected transition probabilities. Then we monotonically update $Q_i(s,a_i)$ by

$$Q_i(s,a_i) = \max \left( Q_i(s,a_i), Q_i^{\text{e}}(s,a_i) \right). \tag{8}$$

We define (7) and (8) together as ***simplified best possible operator***. By performing simplified best possible operator, $Q_i(s,a_i)$ is efficiently updated towards the maximal expected value. And we have the following lemma.

**Lemma 4.** $Q_i(s,a_i)$ *converges to* $\max_{\boldsymbol{a}_{-i}} Q(s,a_i,\boldsymbol{a}_{-i})$ *under simplified best possible operator.*

*Proof.* According to (8), as $Q_i(s,a_i)$ is monotonically increased, $Q_i^k(s,a_i) \geq Q_i^{k-1}(s,a_i)$ in the update $k$. Similar to the proof of Lemma 2, we can easily prove $\max_{\boldsymbol{a}_{-i}} Q(s,a_i,\boldsymbol{a}_{-i}) \geq Q_i^k(s,a_i)$ under (7) and (8). Thus, $\{Q_i^k(s,a_i)\}$ is an increasing sequence and bounded above. According to the monotone convergence theorem, $\{Q_i^k(s,a_i)\}$ converges when $k \to \infty$, and let $Q_i(s,a_i) := Q_i^\infty(s,a_i)$.

Then we prove that the converged value $Q_i(s,a_i)$ is equal to $\max_{\boldsymbol{a}_{-i}} Q(s,a_i,\boldsymbol{a}_{-i})$. Due to monotonicity and convergence, $\forall \epsilon, s, a_i, \exists K$, when $k > K$, $Q_i^k(s,a_i) - Q_i^{k-1}(s,a_i) \leq \epsilon$, no matter which $\tilde{P}_i$ is selected in the update $k$. Since each $\tilde{P}_i$ is possible to be selected, when selecting $\tilde{P}_i(s'|s,a_i) = \arg \max_{P_i(s'|s,a_i)} \mathbb{E}_{P_i(s'|s,a_i)} \left[ r + \gamma \max_{a_i'} Q_i^{k-1}(s',a_i') \right] = P_i^*(s'|s,a_i)$, by performing (7) and (8), we have

$$Q_i^{k-1}(s,a_i) + \epsilon \geq Q_i^k(s,a_i) \geq Q_i^{\text{e}}(s,a_i) = \sum_{s'} P_i^*(s'|s,a_i) \left[ r(s,s') + \gamma \max_{a_i'} Q_i^{k-1}(s',a_i') \right].$$

According to the proof of Lemma 3, we have

$$\max_{s,a_i} \left( \max_{\boldsymbol{a}_{-i}} Q(s,a_i,\boldsymbol{a}_{-i}) - Q_i^{\text{e}}(s,a_i) \right) \leq \gamma \max_{s,a_i} \left( \max_{\boldsymbol{a}_{-i}} Q(s,a_i,\boldsymbol{a}_{-i}) - Q_i^{k-1}(s,a_i) \right).$$

Use $s^*, a_i^*$ to denote

$$\arg\max_{s,a_i} \left( \max_{\boldsymbol{a}_{-i}} Q(s, a_i, \boldsymbol{a}_{-i}) - Q_i^{k-1}(s, a_i) \right).$$

Since $Q_i^{k-1}(s, a_i) + \epsilon \geq Q_i^{\mathrm{e}}(s, a_i)$,

$$\max_{\boldsymbol{a}_{-i}} Q(s^*, a_i^*, \boldsymbol{a}_{-i}) - Q_i^{k-1}(s^*, a_i^*) - \epsilon \leq \gamma\max_{\boldsymbol{a}_{-i}} Q(s^*, a_i^*, \boldsymbol{a}_{-i}) - \gamma Q_i^{k-1}(s^*, a_i^*).$$

Then, we have

$$\left\| \max_{\boldsymbol{a}_{-i}} Q(s, a_i, \boldsymbol{a}_{-i}) - Q_i^{k-1}(s, a_i) \right\|_\infty \leq \frac{\epsilon}{1-\gamma}.$$

Thus, $Q_i(s, a_i)$ converges to $\max_{\boldsymbol{a}_{-i}} Q(s, a_i, \boldsymbol{a}_{-i})$. $\qquad\square$

According to Lemma 1 and 4, we also have:

**Theorem 2.** *The agents learn the optimal joint policy under simplified best possible operator when there is only one optimal joint policy.*

## 2.4 BEST POSSIBLE Q-LEARNING

***Best possible Q-learning*** (BQL) is instantiated on simplified best possible operator. We first consider learning Q-table for tabular cases. The key challenge is how to obtain all possible transition probabilities under deterministic $\boldsymbol{\pi}_{-i}$ during learning. To solve this issue, the whole training process is divided into $M$ epochs. At the epoch $m$, each agent $i$ randomly and independently initializes a deterministic policy $\hat{\pi}_i^m$ and selects a subset of states $S_i^m$. Then each agent $i$ interacts with the environment using the deterministic policy

$$\begin{cases} \arg\max_{a_i} Q_i(s, a_i) & \text{if } s \notin S_i^m, \\ \hat{\pi}_i^m(s) & \text{else.} \end{cases}$$

Each agent $i$ stores independent experiences $(s, a_i, s', r)$ in the replay buffer $\mathcal{D}_i^m$. As $P_i$ depends on $\boldsymbol{\pi}_{-i}$ and agents act deterministic policies, $\mathcal{D}_i^m$ contains one $P_i$ under a deterministic $\boldsymbol{\pi}_{-i}$. Since $P_i$ will change if other agents modify their policies $\boldsymbol{\pi}_{-i}$, acting the randomly initialized policy $\hat{\pi}_i^m$ on $S_i^m$ in the epoch $m$ not only helps each agent $i$ to explore state-action pairs, but also helps other agents to explore possible transition probabilities. When $M$ is sufficiently large, given any $(s, a_i)$ pair, any $P_i(s, a_i)$ can be found in a replay buffer.

After interaction of the epoch $m$, each agent $i$ has a buffer series $\{\mathcal{D}_i^1, \cdots, \mathcal{D}_i^m\}$, each of which has different transition probabilities. At training period of the epoch $m$, each agent $i$ randomly selects one replay buffer $\mathcal{D}_i^j$ from $\{\mathcal{D}_i^1, \cdots, \mathcal{D}_i^m\}$ and samples mini-batches $\{s, a_i, s', r\}$ from $\mathcal{D}_i^j$ to update Q-table $Q_i^{\mathrm{e}}(s, a_i)$ by (7), and then samples mini-batches from $\mathcal{D}_i^j$ to update $Q_i(s, a_i)$ by (8). The Q-table implementation is summarized in Algorithm 1.

The sample efficiency of collecting the buffer series seems to be a limitation of BQL, and we further analyze it. Simplified best possible operator requires that any possible $P_i(s, a_i)$ of $(s, a_i)$ pair can be found in one buffer, but does not care about the relationship between transition probabilities of different state-action pairs in the same buffer. So BQL ideally needs only $|\mathcal{A}_i| \times |\mathcal{A}_{-i}| = |\mathcal{A}|$ small buffers to cover all possible $P_i$ for any $(s, a_i)$ pair, which is very efficient for experience collection. We give an intuitive illustration for this and analyze that BQL has similar sample complexity to the joint Q-learning (3) in Appendix B.

In complex environments with large or continuous state-action space, it is inefficient and costly to follow the experience collection in tabular cases, where the agents cannot update their policies during the interaction of each epoch and each epoch requires adequate samples to accurately estimate the expectation (7). Thus, in complex environments, same as IQL, each agent $i$ only maintains one replay buffer $\mathcal{D}_i$, which contains all historical experiences, and uses the same $\epsilon$-greedy policy as IQL (without the randomly initialized deterministic policy $\hat{\pi}_i$). Then we instantiate simplified best possible operator with neural networks $Q_i$ and $Q_i^{\mathrm{e}}$. $Q_i^{\mathrm{e}}$ is updated by minimizing:

$$\mathbb{E}_{s,a_i,s',r\sim\mathcal{D}_i} \left[ (Q_i^{\mathrm{e}}(s, a_i) - r - \gamma Q_i(s', a_i'^*))^2 \right], \quad a_i'^* = \arg\max_{a_i'} Q_i(s', a_i'). \qquad (9)$$

---

**Algorithm 1** BQL with Q-table for each agent $i$

---

1: Initialize tables $Q_i$ and $Q_i^e$.
2: **for** $m = 1, \ldots, M$ **do**
3:     Initialize the replay buffer $\mathcal{D}_i^m$ and the exploration policy $\hat{\pi}_i^m$.
4:     All agents interact with the environment and store experiences $(s, a_i, s', r)$ in $\mathcal{D}_i^m$.
5:     **for** $t = 1, \ldots, n\_update$ **do**
6:         Randomly select a buffer $\mathcal{D}_i^j$ from $\mathcal{D}_i^1, \cdots, \mathcal{D}_i^m$.
7:         Update $Q_i^e$ according to (7) by sampling from $\mathcal{D}_i^j$.
8:         Update $Q_i$ according to (8) by sampling from $\mathcal{D}_i^j$.
9:     **end for**
10: **end for**

---

**Algorithm 2** BQL with neural network for each agent $i$

---

1: Initialize neural networks $Q_i$ and $Q_i^e$, and the target network $\bar{Q}_i^e$.
2: Initialize the replay buffer $\mathcal{D}_i$.
3: **for** $t = 1, \ldots, n\_iteration$ **do**
4:     All agents interact with the environment and store experiences $(s, a_i, s', r)$ in $\mathcal{D}_i$.
5:     Sample a mini-batch from $\mathcal{D}_i$.
6:     Update $Q_i^e$ by minimizing (9).
7:     Update $Q_i$ by minimizing (10).
8:     Update the target networks $\bar{Q}_i^e$.
9: **end for**

---

And $Q_i$ is updated by minimizing:

$$\mathbb{E}_{s,a_i \sim \mathcal{D}_i} \left[ w(s, a_i) \left( Q_i(s, a_i) - \bar{Q}_i^e(s, a_i) \right)^2 \right], \quad w(s, a_i) = \begin{cases} 1 & \text{if } \bar{Q}_i^e(s, a_i) > Q_i(s, a_i) \\ \lambda & \text{else.} \end{cases} \quad (10)$$

$\bar{Q}_i^e$ is the softly updated target network of $Q_i^e$. When $\lambda = 0$, (10) is equivalent to (8). However, when $\lambda = 0$, the positive random noise of $Q_i$ in the update can be continuously accumulated, which may cause value overestimation. So we adopt the weighted max in (10) by setting $0 < \lambda < 1$ to offset the positive random noise. In continuous action space, following DDPG (Lillicrap et al., 2016), we train a policy network $\pi_i(s)$ by maximizing $Q_i(s, \pi_i(s))$ as a substitute of $\arg\max_{a_i} Q_i(s, a_i)$. The neural network implementation is summarized in Algorithm 2.

Simplified best possible operator is meaningful for neural network implementation. As there is only one buffer $\mathcal{D}_i$, we cannot perform (6) but can still perform (7) and (8) on $\mathcal{D}_i$. As other agents are updating their policies, the transition probabilities in $\mathcal{D}_i$ will continuously change. If $\mathcal{D}_i$ sufficiently goes through all possible transition probabilities, $Q_i(s, a_i)$ converges to $\max_{\boldsymbol{a}_{-i}} Q(s, a_i, \boldsymbol{a}_{-i})$ and the agents learn the optimal joint policy. That is to say, *non-stationarity in the replay buffer is no longer a drawback, but a necessary condition for BQL.*

## 3 RELATED WORK

Most existing MARL methods (Lowe et al., 2017; Iqbal & Sha, 2019; Wang et al., 2020; Zhang et al., 2021b; Su & Lu, 2022; Peng et al., 2021; Li et al., 2022; Sunehag et al., 2018; Rashid et al., 2018; Son et al., 2019) follow centralized training and decentralized execution (CTDE), where the information of all agents can be accessed in a centralized way during training. Unlike these methods, we focus on fully decentralized learning where global information is not available. The most straightforward decentralized methods, *i.e.*, independent Q-learning (Tan, 1993) and independent PPO (IPPO) (de Witt et al., 2020a), cannot guarantee the convergence of the learned policy, because the transition probabilities are non-stationary from the perspective of each agent as all agents are learning policies simultaneously. Multi-agent alternate Q-learning (MA2QL) (Su et al., 2022) guarantees the convergence to a Nash equilibrium, but the converged equilibrium may not be the optimal one when there are multiple Nash equilibria. Moreover, to obtain the theoretical guarantee, it has to be trained in an on-policy manner and cannot use replay buffers, which leads to poor sample efficiency. Following the principle of optimistic estimation, Hysteretic IQL (Matignon et al., 2007) sets

a slow learning rate to the value punishment. Distributed IQL (Lauer & Riedmiller, 2000), a special case of Hysteretic IQL with the slow learning rate being zero, guarantees the convergence to the optimum but only in deterministic environments. I2Q (Jiang & Lu, 2022) lets each agent perform independent Q-learning on ideal transition probabilities and could learn the optimal policy only in deterministic environments. Our BQL is the first fully decentralized algorithm that converges to the optimal joint policy in stochastic environments.

In the next section, we compare BQL against these Q-learning variants (Distributed IQL is included in Hysteretic IQL). Comparing with on-policy algorithms, *e.g.*, IPPO, that are not sample-efficient especially in fully decentralized settings, is out of focus and thus deferred to Appendix. Decentralized methods with communication (Zhang et al., 2018; Konan et al., 2021; Li & He, 2020) allow information sharing with neighboring agents according to a communication channel. However, they do not follow the fully decentralized setting and thus are beyond the scope of this paper.

## 4 EXPERIMENTS

We first test BQL with Q-table on randomly generated cooperative stochastic games to verify its convergence and optimality. Then, to illustrate its performance on complex tasks, we compare BQL with neural networks against Q-learning variants on MPE-version differential games (Jiang & Lu, 2022), Multi-Agent MuJoCo (Peng et al., 2021), SMAC (Samvelyan et al., 2019), and GRF (Kurach et al., 2020). The experiments cover both fully and partially observable, deterministic and stochastic, discrete and continuous environments. Since we consider the fully decentralized setting, BQL and the baselines *do not use parameter sharing*. The results are presented using mean and standard. More details about hyperparameters are available in Appendix E.

### 4.1 STOCHASTIC GAMES

To support the theoretical analysis of BQL, we test the Q-table instantiation on stochastic games with 4 agents, 30 states, and infinite horizon. The action space of each agent is 4, so the joint action space $|\mathcal{A}| = 256$. The distribution of initial states is uniform. Each state will transition to any state given a joint action according to transition probabilities. The transition probabilities and reward function are randomly generated and fixed in each game. We randomly generate 20 games and train the agents for four different seeds in each game.

The mean normalized return and std over the 20 games are shown in Figure 1a. IQL cannot learn the optimal policies due to non-stationarity. Although using the optimistic update to remedy the non-stationarity, Hysteretic IQL (H-IQL) still cannot solve this problem in stochastic environments and shows similar performance to IQL. In Appendix A, we thoroughly analyze the difference and relationship between H-IQL and BQL. I2Q performs Q-learning on the ideal transition function where the next state is deterministically the one with the highest value, which however is impossible in stochastic tasks. So I2Q cannot guarantee the optimal joint policy in stochastic environments. MA2QL guarantees the convergence to a Nash equilibrium, but the converged one may not be the optimal one, thus there is a performance gap between MA2QL and optimal policies. BQL could converge to the optimum, and the tiny gap is caused by the fitting error of the Q-table update. This verifies our theoretical analysis. Note that, in Q-table instantiations, MA2QL and BQL use different experience collection from IQL, *i.e.*, exploration strategy and replay buffer. MA2QL only uses on-policy experiences and BQL collects a series of small buffers. However, *for sample efficiency, the two methods have to use the same experience collection as IQL in complex tasks with neural networks.* MA2QL- and BQL- respectively denote the two methods with the same experience collection as IQL. Trained on off-policy experiences, MA2QL- suffers from non-stationarity and achieves similar performance to IQL. Even if using only one buffer, as we have analyzed in Section 2.4, if the non-stationary buffer sufficiently goes through all possible transition probabilities, BQL agents can also converge to the optimum. Although going through all possible transition probabilities by one buffer is inefficient, BQL- significantly outperforms IQL, which implies the potential of BQL with one buffer in complex tasks.

Figure 1b shows the effect of the size of buffer $\mathcal{D}_i^m$ at the epoch $m$. If $|\mathcal{D}_i^m|$ is too small, *i.e.*, 200, the experiences in $|\mathcal{D}_i^m|$ are insufficient to accurately estimate the expected value (7). If $|\mathcal{D}_i^m|$ is too large, *i.e.*, 10000, the experiences in $|\mathcal{D}_i^m|$ are redundant, and the buffer series is difficult to cover

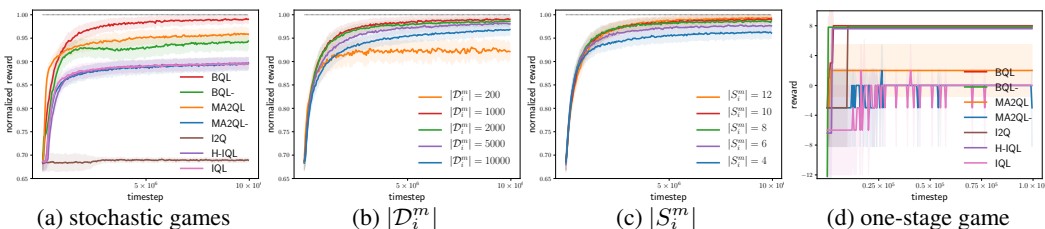

Figure 1: Learning curves on cooperative stochastic games (normalized by the optimal return).

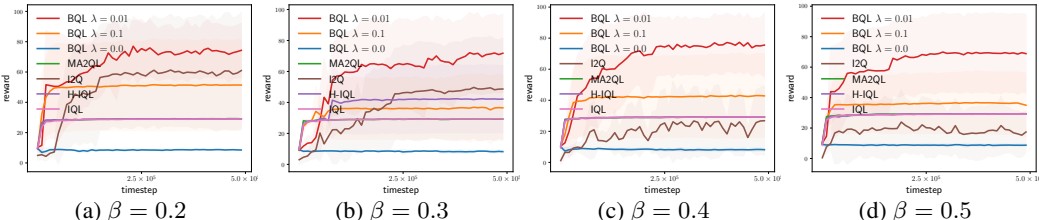

Figure 2: Learning curves on MPE-based differential games with different $\beta$.

all possible transition probabilities given fixed total training timesteps. Figure 1c shows the effect of the number of states on which the agents perform the randomly initialized deterministic policy $\hat{\pi}_i^m$ for exploration. The larger $|S_i^m|$ means a stronger exploration for both state-action pairs and possible transition probabilities, which leads to better performance.

We then consider a one-stage game that is wildly adopted in MARL (Son et al., 2019). There are 2 agents, and the action space of each agent is 3. The reward matrix is

$$
\begin{vmatrix}
a_1/a_2 & \mathcal{A}^{(1)} & \mathcal{A}^{(2)} & \mathcal{A}^{(3)} \\
\mathcal{A}^{(1)} & \mathbf{8} & -12 & -12 \\
\mathcal{A}^{(2)} & -12 & 0 & 0 \\
\mathcal{A}^{(3)} & -12 & 0 & 0
\end{vmatrix}
$$

where the reward 8 is the global optimum and the reward 0 is the sub-optimal Nash equilibrium. As shown in Figure 1d, MA2QL converges to the sub-optimal Nash equilibrium when the initial policy of the second agent selects $\mathcal{A}^{(2)}$ or $\mathcal{A}^{(3)}$. But BQL converges to the global optimum easily.

### 4.2 MPE

To evaluate the effectiveness of BQL with neural network implementation, we adopt the 3-agent MPE-based differential game used in I2Q (Jiang & Lu, 2022), where 3 agents can move in the range $[-1, 1]$. Different from the original deterministic version, we add stochasticity to it. In each timestep, agent $i$ acts the action $a_i \in [-1, 1]$, and the position of agent $i$ will be updated as $x_i = \text{clip}(x_i + 0.1 \times a_i, -1, 1)$ (i.e., the updated position is clipped to $[-1, 1]$) with the probability $1 - \beta$, or will be updated as $-x_i$ with the probability $\beta$. $\beta$ controls the stochasticity. The state is the vector of positions $\{x_1, x_2, x_3\}$. The reward function of each timestep is

$$
r = \begin{cases}
0.5 \cos(4l\pi) + 0.5 & \text{if } l \leq 0.25 \\
0 & \text{if } 0.25 < l \leq 0.6 \\
0.15 \cos(5\pi(l - 0.8)) + 0.15 & \text{if } 0.6 < l \leq 1.0 \\
0 & \text{if } l > 1.0
\end{cases}, \quad l = \sqrt{\frac{2}{3}(x_1^2 + x_2^2 + x_3^2)}.
$$

We visualize the relation between $r$ and $l$ in Figure 12. There is only one global optimum ($l = 0$ and $r = 1$) but infinite sub-optima ($l = 0.8$ and $r = 0.3$), and the narrow region with $r > 0.3$ is surrounded by the region with $r = 0$. So it is quite a challenge to learn the optimal policies in a fully decentralized way. Each episode contains 100 timesteps, and the initial positions follow the uniform distribution. We perform experiments with different stochasticities $\beta$, and train the agents for eight seeds with each $\beta$. In continuous environments, BQL and baselines are built on DDPG.

As shown in Figure 2, IQL always falls into the local optimum (total reward $\approx 30$) because of the non-stationary transition probabilities. H-IQL only escapes the local optimum in one seed in the setting with $\beta = 0.3$. According to the theoretical analysis in I2Q paper, the value estimation

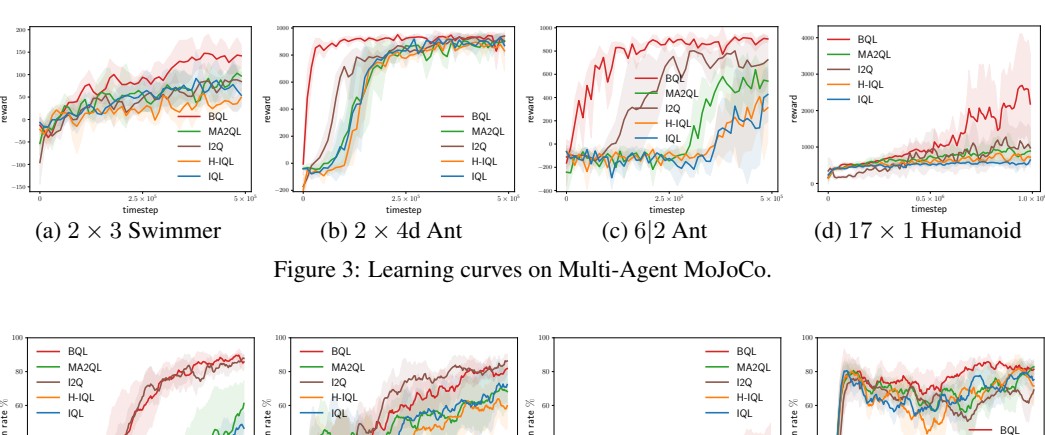

Figure 3: Learning curves on Multi-Agent MoJoCo.

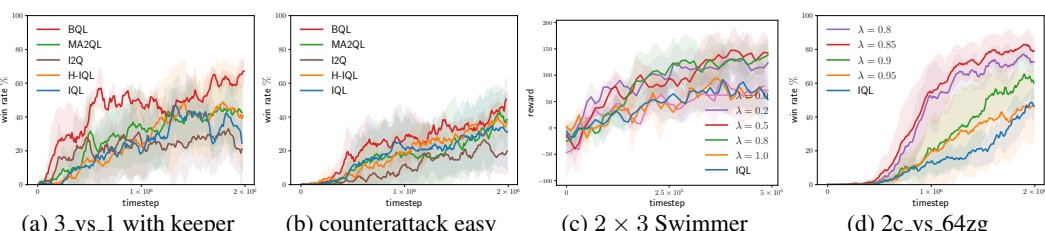

Figure 4: Learning curves on SMAC.

Figure 5: (a) and (b): Learning curves on GRF. (c) and (d): Learning curves with different $\lambda$.

error of I2Q will become larger when stochasticity grows, which is the reason why I2Q shows poor performance with $\beta = 0.4$ and $0.5$. In neural network implementations, MA2QL and BQL use the same experience collection as IQL, so there is no MA2QL- and BQL-. MA2QL converges to the local optimum because it cannot guarantee that the converged equilibrium is the global optimum, especially trained using off-policy data. BQL ($\lambda = 0.01$) can escape from local optimum in more than 4 seeds in all settings, which demonstrates the effectiveness of our optimization objectives (9) and (10). The difference between global optimum (total reward $\approx 100$) and local optimum is large, which results in the large variance of BQL. In the objective (10), $\lambda$ controls the balance between performing best possible operator and offsetting the overestimation caused by the operator. As shown in Figure 2, the large $\lambda$, *i.e.*, 0.1, will weaken the strength of BQL, while too small $\lambda$, *i.e.*, 0, will cause severe overestimation and destroy the performance.

### 4.3 MULTI-AGENT MUJOCO

To evaluate BQL in *partially observable* environments, we adopt Multi-Agent MuJoCo (Peng et al., 2021), where each agent independently controls one or some joints of the robot. In each task, we test four random seeds and plot the learning curves in Figure 3. Here, we set $\lambda = 0.5$. In the first three tasks, each agent can only observe the state of its own joints and bodies (with the parameter agent_obsk = 0). BQL achieves higher reward or learns faster than the baselines, which verifies that BQL could be applied to partially observable environments.

In the first three tasks, we only consider two-agent cases in the partially observable setting, because the too limited observation range cannot support strong policies when there are more agents. We also test BQL on 17-agent Humanoid with full observation in Figure 3d. BQL obtains significant performance gain in this many-agent task, which can be evidence of the **good scalability** of BQL.

### 4.4 SMAC AND GOOGLE RESEARCH FOOTBALL

We also perform experiments on *partially observable and stochastic* SMAC tasks (Samvelyan et al., 2019) with the version SC2.4.10, including both easy and hard maps (Yu et al., 2021). Agent numbers vary between 2 and 9. We build BQL on the implementation of PyMARL (Samvelyan et al., 2019) and train the agents for four random seeds. The learning curves are shown in Figure 4. In general, BQL outperforms the baselines, which verifies that BQL can also obtain performance gain in high-dimensional complex tasks. In 2c_vs_64zg, by considering the non-stationary transition probabilities, BQL and I2Q achieve significant improvement over other methods. We conjecture that the interplay between agents is strong in this task.

GRF (Kurach et al., 2020) is a physics-based 3D simulator where agents aim to master playing football. We select two academy tasks with sparse rewards: 3_vs_1 with keeper (3 agents) and counterattack easy (4 agents). We build BQL on the implementation of PyMARL2 (Hu et al., 2021) and train the agents for four random seeds. Although I2Q shows similar results with BQL in some SMAC tasks, BQL can outperform I2Q in GRF as shown in Figure 5a and 5b, because GRF is more stochastic than SMAC and the value gap of I2Q will enlarge along with the increase of stochasticity.

### 4.5 HYPERPARAMETER $\lambda$

We further investigate the effectiveness of $\lambda$ in Multi-Agent MuJoCo and SMAC. In the objective (10), $\lambda$ controls the balance between performing best possible operator and offsetting the overestimation caused by the operator. As shown in Figure 5c and 5d, too large $\lambda$ will weaken the strength of BQL. When $\lambda = 1.0$, BQL degenerates into IQL. Too small $\lambda$, *i.e.*, 0, will cause overestimation. If the environment is more complex, *e.g.*, SMAC, overestimation is more likely to occur, so we should set a large $\lambda$. In $2 \times 3$ Swimmer, when $\lambda$ falls within the interval $[0.2, 0.8]$, BQL can obtain performance gain, showing the robustness to $\lambda$.

## 5 CONCLUSION

We propose *best possible operator* and theoretically prove that the policies of agents will converge to the optimal joint policy if each agent independently updates its individual state-action value by the operator. We then simplify the operator and derive BQL, the first decentralized MARL algorithm that guarantees the convergence to the global optimum in stochastic environments. Empirically, BQL outperforms baselines in a variety of multi-agent tasks. We also discuss the limitation of unique optimal joint policy and sample efficiency, and provide corresponding solutions for BQL.

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

# A  COMPARISON WITH HYSTERETIC IQL

Hysteretic IQL is a special case of BQL when the environment is deterministic. To thoroughly illustrate that, we rewrite the loss function of BQL

$$w(s, a_i) \left( Q_i\left(s, a_i\right) - \mathbb{E}_{\tilde{P}_i(s'|s, a_i)} \left[ r + \gamma \max_{a_i'} Q_i(s', a_i') \right] \right)^2,$$

$$w(s, a_i) = \begin{cases} 1 & \text{if } \mathbb{E}_{\tilde{P}_i(s'|s, a_i)} \left[ r + \gamma \max_{a_i'} Q_i(s', a_i') \right] > Q_i\left(s, a_i\right) \\ \lambda & \text{else.} \end{cases}$$

If $\lambda = 0$, the update of BQL is

$$Q_i(s, a_i) = \max \left( Q_i(s, a_i), \mathbb{E}_{\tilde{P}_i(s'|s, a_i)} \left[ r + \gamma \max_{a_i'} Q_i(s', a_i') \right] \right).$$

Hysteretic IQL follows the loss function

$$w(s, a_i) \left( Q_i\left(s, a_i\right) - r - \gamma \max_{a_i'} Q_i(s', a_i') \right)^2,$$

$$w(s, a_i) = \begin{cases} 1 & \text{if } r + \gamma \max_{a_i'} Q_i(s', a_i') > Q_i\left(s, a_i\right) \\ \lambda & \text{else.} \end{cases}$$

If $\lambda = 0$, Hysteretic IQL degenerates into Distributed IQL (Lauer & Riedmiller, 2000)

$$Q_i(s, a_i) = \max \left( Q_i(s, a_i), r + \gamma \max_{a_i'} Q_i(s', a_i') \right).$$

BQL takes the max of the expected target on transition probability $\tilde{P}_i(s'|s, a_i)$, while Hysteretic IQL takes the max of the target on the next state $s'$. When the environment is deterministic, they are equivalent. However, in stochastic environments, Hysteretic IQL cannot guarantee to converge to the global optimum since the environment will not always transition to the same $s'$. BQL can guarantee the global optimum in both deterministic and stochastic environments.

# B  EFFICIENCY OF BQL

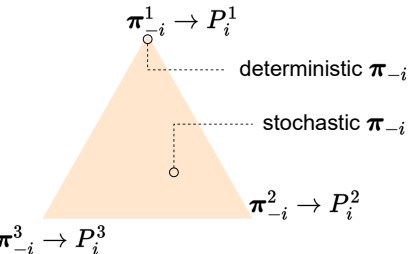

Figure 6: Space of other agents' policies $\boldsymbol{\pi}_{-i}$ given an $(s, a_i)$.

We will discuss the efficiency of collecting the replay buffer for BQL. The space of other agents' policies $\boldsymbol{\pi}_{-i}$ given $(s, a_i)$ pair is a convex polytope. For clarity, Figure 6 shows a triangle space. Each $\boldsymbol{\pi}_{-i}$ corresponds to a $P_i(s'|s, a_i)$. Deterministic policies $\boldsymbol{\pi}_{-i}$ locate at the vertexes, while the edges and the inside of the polytope are stochastic $\boldsymbol{\pi}_{-i}$, the mix of deterministic ones. Since BQL only considers deterministic policies, the buffer series only needs to cover all the vertexes by acting deterministic policies in the collection of each buffer $\mathcal{D}_i^m$, which is efficient. BQL needs only $|\mathcal{A}_i| \times |\mathcal{A}_{-i}| = |\mathcal{A}|$ small buffers, which is irrelevant to state space $|\mathcal{S}|$, to meet the requirement of simplified best possible operator that any one of possible $P_i(s'|s, a_i)$ can be found in one (ideally only one) buffer given $(s, a_i)$ pair. More specifically, $|\mathcal{A}_i|$ buffers are needed to cover action space, and $|\mathcal{A}_{-i}|$ buffers are needed to cover transition space for each action. We intuitively illustrate this in Figure 7. Each state in $\mathcal{D}_i^m$ requires # samples to estimate the expectation in (7), so the

Ideally 4 buffers cover all possible $P_i(s, a_i)$

$$< s^1, a_i^1, P_i^1(s^1, a_i^1) >, < s^2, a_i^1, P_i^1(s^2, a_i^1) >, < s^3, a_i^1, P_i^1(s^3, a_i^1) >$$

$$< s^1, a_i^1, P_i^2(s^1, a_i^1) >, < s^2, a_i^1, P_i^2(s^2, a_i^1) >, < s^3, a_i^1, P_i^2(s^3, a_i^1) >$$

$$< s^1, a_i^2, P_i^1(s^1, a_i^2) >, < s^2, a_i^2, P_i^1(s^2, a_i^2) >, < s^3, a_i^2, P_i^1(s^3, a_i^2) >$$

$$< s^1, a_i^2, P_i^2(s^1, a_i^2) >, < s^2, a_i^2, P_i^2(s^2, a_i^2) >, < s^3, a_i^2, P_i^2(s^3, a_i^2) >$$

Figure 7: Toy case for illustrating the ideal buffer number. $|\mathcal{S}| = 3$, $|\mathcal{A}_i| = 2$, and $|\mathcal{A}_{-i}| = 2$ corresponding to $P_i^1$ and $P_i^2$. We can see that any $P_i(s, a_i)$ can be found in the 4 buffers.

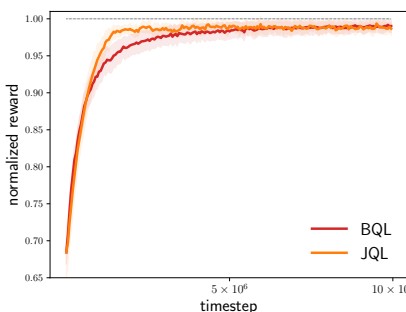

Figure 8: Learning curves of BQL and joint Q-learning (JQL). BQL shows similar sample efficiency to JQL.

sample complexity is $O(|\mathcal{A}||S|\#)$. For the joint Q-learning (3), the most efficient known method to guarantee the convergence and optimality in stochastic environments, each state-joint action pair $(s, \boldsymbol{a})$ requires $\#$ samples to estimate the expectation, so the sample complexity is also $O(|\mathcal{A}||S|\#)$. Thus, BQL is close to the joint Q-learning in terms of sample complexity, which is empirically verified in Figure 8.

One may ask "since you obtain all possible transition probabilities, why not perform IQL on each transition probability and choose the highest value?" Actually, this naive algorithm can also learn the optimal policy, but the buffer collection of the naive algorithm is much more costly than that of BQL. The naive algorithm requires that any one of possible *transition probability functions of the whole state-action space* could be found in one buffer, which needs $|\mathcal{A}_{-i}|^{|\mathcal{S}|}$ buffers. And training IQL $|\mathcal{A}_{-i}|^{|\mathcal{S}|}$ times is also formidable. BQL only requires that any one of possible *transition probability of any state-action pair* could be found in one buffer, which is much more efficient.

However, considering sample efficiency, BQL with neural networks only maintains one replay buffer $\mathcal{D}_i$ containing all historical experiences, which is the same as IQL. $P_i$ in $\mathcal{D}_i$ corresponds to the average of other agents' historical policies, which is stochastic. Therefore, to guarantee the optimality, in theory, BQL with one buffer has to go through almost the whole $\boldsymbol{\pi}_{-i}$ space, which is costly. As shown in Figure 1d, BQL- (with one buffer) outperforms IQL but cannot achieve similar results as BQL (with buffer series), showing that maintaining one buffer is costly but still effective. In neural network instantiation, we show the results of BQL with the buffer series in Figure 9. Due to sample efficiency, the buffer series cannot achieve strong performance, and maintaining one buffer like IQL is a better choice in complex environments.

## C   OTHER BASE ALGORITHMS

Besides DDPG, BQL could also be built on other variants of Q-learning, *e.g.*, SAC. Figure 10 shows that BQL could also obtain performance gain on independent SAC. Independent PPO (IPPO) (de Witt et al., 2020a) is an on-policy decentralized MARL baseline. IPPO is not a Q-learning method so it cannot be the base algorithm of BQL. On-policy algorithms do not use old experiences,

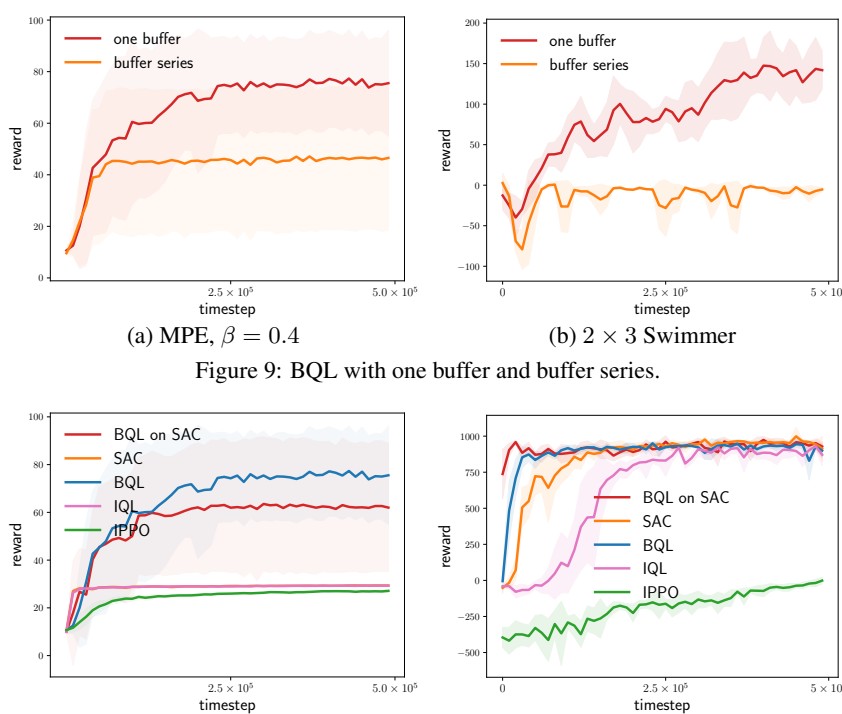

(a) MPE, $\beta = 0.4$         (b) $2 \times 3$ Swimmer

Figure 9: BQL with one buffer and buffer series.

(a) MPE, $\beta = 0.4$         (b) $2 \times 4$d Ant

Figure 10: Learning curves of other base algorithms.

which makes them weak on sample efficiency (Achiam, 2018) especially in fully decentralized settings as shown in Figure 10. Thus, it is unfair to compare off-policy algorithms with on-policy algorithms.

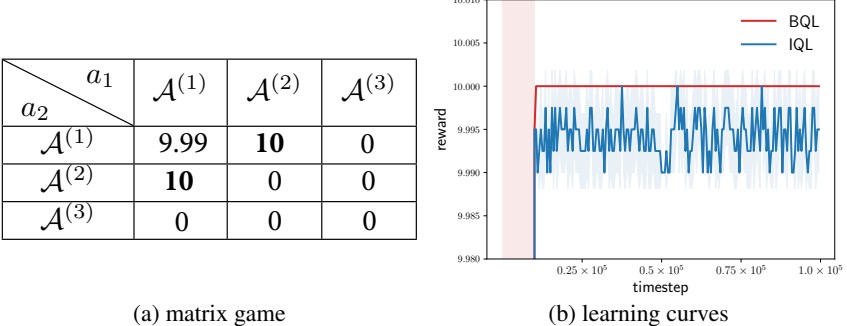

| $a_1$ $a_2$ | $\mathcal{A}^{(1)}$ | $\mathcal{A}^{(2)}$ | $\mathcal{A}^{(3)}$ |
|---|---|---|---|
| $\mathcal{A}^{(1)}$ | 9.99 | **10** | 0 |
| $\mathcal{A}^{(2)}$ | **10** | 0 | 0 |
| $\mathcal{A}^{(3)}$ | 0 | 0 | 0 |

(a) matrix game         (b) learning curves

Figure 11: Learning curves on a one-stage matrix game with multiple optimal joint policies.

## D  MULTIPLE OPTIMAL JOINT POLICIES

We assume that there is only one optimal joint policy. With multiple optimal actions (with the max $Q_i(s, a_i)$), if each agent arbitrarily selects one of the optimal independent actions, the joint action might not be optimal. To address this, we use the simple technique proposed in I2Q (Jiang & Lu, 2022). Concretely, we set a performance tolerance $\varepsilon$ and introduce a fixed randomly initialized reward function $\hat{r}(s, s') \in (0, (1 - \gamma)\varepsilon]$. Then all agents perform BQL to learn $\hat{Q}_i(s, a_i)$ of the shaped reward $r + \hat{r}$. Since $\hat{r} > 0$, $\hat{Q}_i(s, a_i) > Q_i(s, a_i)$. In $\hat{Q}_i(s, a_i)$, the maximal contribution from $\hat{r}$ is $(1 - \gamma)\varepsilon / (1 - \gamma) = \varepsilon$, so the minimal contribution from $r$ is $\hat{Q}_i(s, a_i) - \varepsilon > Q_i(s, a_i) - \varepsilon$, which means that the maximal performance drop is $\varepsilon$ when selecting actions according to $\hat{Q}_i$. It is a small probability event to find multiple optimal joint policies on the reward function $r + \hat{r}$, because $\hat{r}(s, s')$ is randomly initialized. Thus, if $\varepsilon$ is set to be small enough, BQL can solve the task with

multiple optimal joint policies. However, this technique is introduced to only remedy the assumption for theoretical results. Empirically, this is not required, because there is usually only one optimal joint policy in complex environments. In all experiments, we do not use the randomly initialized reward function for BQL and other baselines, so the comparison is fair.

We test the randomly initialized reward function on a one-stage matrix game with two optimal joint policies $(1, 2)$ and $(2, 1)$, as shown in Figure 11. If the agents independently select actions, they might choose the miscoordinated joint policies $(1, 1)$ and $(2, 2)$. IQL cannot converge, but BQL agents always select coordinated actions, though the value gap between the optimal policy and suboptimal policy is so small, which verifies the effectiveness of the randomly initialized reward.

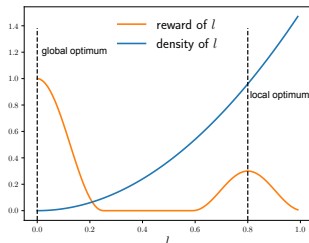

Figure 12: Curves of reward and density of $l = \sqrt{\frac{2}{3} \sum_{i=0}^{3} x_i^2}$ in MPE. We plot the density of uniform state distribution. There is only one global optimum, but the density of local optimum is high. So decentralized agents will easily learn the local optimal policies.

## E  HYPERPARAMETERS

In MPE-based (MIT license) differential games, the relationship between $r$ and $l$ is visualized in Figure 12.

In $2 \times 3$ Swimmer, there are two agents and each of them controls 3 joints of ManyAgent Swimmer. In $6|2$ Ant, there are two agents. One of them controls 6 joints, and one of them controls 2 joints. And so on.

In MPE-based differential games and Multi-Agent MuJoCo (MIT license), we adopt SpinningUp (Achiam, 2018) implementation (MIT license), the SOTA implementation of DDPG, and follow all hyperparameters in SpinningUp. The discount factor $\gamma = 0.99$, the learning rate is $0.001$ with Adam optimizer, the batch size is $100$, the replay buffer contains $5 \times 10^5$ transitions, the hidden units are $256$.

In SMAC (MIT license), we adopt PyMARL (Samvelyan et al., 2019) implementation and follow all hyperparameters in PyMARL (Apache-2.0 license). The discount factor $\gamma = 0.99$, the learning rate is $0.0005$ with RMSprop optimizer, the batch size is $32$ episodes, the replay buffer contains $5000$ episodes, the hidden units are $64$. We adopt the version SC2.4.10 of SMAC.

In GRF (Apache-2.0 license), we adopt PyMARL2 (Hu et al., 2021) implementation (Apache-2.0 license) and follow all hyperparameters in PyMARL2. The discount factor $\gamma = 0.999$, the learning rate is $0.0005$ with Adam optimizer, the batch size is $128$ episodes, the replay buffer contains $2000$ episodes, the hidden units are $256$. We use simple115 feature (a 115-dimensional vector summarizing many aspects of the game) as observation instead of RGB image.

In MPE-based differential games, we set $\lambda = 0.01$. In Multi-Agent MuJoCo, we set $\lambda = 0.5$, and in SMAC, we set $\lambda = 0.85$ for 2c_vs_64zg and $\lambda = 0.8$ for other tasks. In GRF, we set $\lambda = 0.1$ for 3_vs_1 with keeper and $\lambda = 0.4$ for counterattack easy.

The experiments are carried out on Intel i7-8700 CPU and NVIDIA GTX 1080Ti GPU. The training of each MPE, MuJoCo, and GRF task could be finished in 5 hours, and the training of each SMAC task could be finished in 20 hours.

