# OpenReview forum: "Best Possible Q-Learning"
_ICLR.cc/2025/Conference — Submitted to ICLR 2025_

### Official Review · Reviewer_bFNq · 2024-10-15

**Soundness:** 2
**Presentation:** 2
**Contribution:** 2
**Rating:** 3
**Confidence:** 3

**Summary:**

This paper studies multi agent RL. A best possible operator is proposed to learn the optimal joint Q function. To address the computational cost, a simplied randomized operator is proposed. Algorithm based on it is further designed and extensive experiments are developed.

**Strengths:**

1. The problem of MARL is definitely important. This work provide a theoretical study on it with convergence guarantees.
2. I do appreciate the experiment part.

**Weaknesses:**

1. The observation of the global state can be a strong assumption.
2. The uniqueness of the joint optimal policy can also be strong.
3. Since I am not familar with MARL, some statements/claims will benifit from more explanations.
4. Some proofs are not convinced.

**Questions:**

1. How is eq (4) obtained? Eq(3) holds when $a=(a_1^*,a_2^*,...,a^*_n)=\pi^*(s)$. Now if taking the maximum of other actions, will it require to fix $a_i=a^*_i$?
2. In the case of there exists onle one optimal policy, why is it deterministic? The Puterman's book is too large and more specific reference should be provided. It also seems to me that the Puterman's book is for single agent case mostly.
3. In the proof of Lemma 4, the convergence is based on the fact that the optimal kernel is chosen during updating. What if the kernel is never choosen?
4. When only deterministic policies are considered, the set of all possible kernels is finite, making Q3 a bit reasonable. But what if random policies are also considered?

---

> ### Author Response · Authors · 2024-11-14
>
> > The observation of the global state can be a strong assumption.
>
> To the best of our knowledge, BQL is the first method to guarantee the convergence and optimality on the global state in decentralized MARL, other baselines even cannot guarantee the optimality on the global state.
>
> > The uniqueness of the joint optimal policy can also be strong.
>
> Whether the optimality can be guaranteed with multiple optimal policies in decentralized MARL is still an open problem. To the best of our knowledge, no method can theoretically solve this problem in fully decentralized settings. In Appendix D, we have discussed how to deal with the case with multiple optimal policies.
>
> > How is eq (4) obtained?
>
> Eq(3) does not require $a = \pi^*(s)$, but describes the optimal value given any $a$. (4) is from taking $\max_{a_{-i}}$ on both sides of (3) so it does not require $a_i = a_i^*$.
>
> > In the case of there exists only one optimal policy, why is it deterministic?
>
> Puterman's book says that there exists at least one deterministic optimal policy in an MDP, so if there exists only one optimal policy, this policy must be deterministic.
>
> > In the proof of Lemma 4, the convergence is based on the fact that the optimal kernel is chosen during updating. What if the kernel is never chosen?
>
> In tabular cases, it is easy to sample all kernels. Any kernel can be chosen so the optimal kernel will be chosen.
>
> > When only deterministic policies are considered, the set of all possible kernels is finite, making Q3 a bit reasonable. But what if random policies are also considered?
>
> There exists at least one deterministic optimal policy in an MDP, so we only need to consider the deterministic policies. Especially when there is only a unique optimal policy, stochastic policies must not be optimal.

---

> > ### Comment · Reviewer_bFNq · 2024-11-20
> >
> > Thank you for your response.
> >
> > Regrading Q2, I am still unclear about how to use the result for single agent to conclude results for multi-agent. Are you treating a multi-agent system as a single one, by considering the global state $s= (s_1,...,s_N)$ and global action $a=(a_1,...,a_N)$?
> >
> >
> > Regrading Q3, the authors claim that 'In tabular cases, it is easy to sample all kernels. Any kernel can be chosen so the optimal kernel will be chosen.' However, shouldn't this exactly be the case that the alforithm try to avoid? As in Lines 181-184, going over all kernels can be too costly, thus the simplified operator is proposed, yet now this operator is only effective under small-scale tabular case.
> >
> > In the case of infinite many kernels exist, the probability of selecting the best one can be 0. I agree the algorithm will converge (increasing bounded sequence will have a limit), but this convergence is only asymptotic which may take infinite time.

---

> ### Author Response · Authors · 2024-11-18
>
> Dear Reviewer bFNq,
>
> Have we addressed your concerns? If you have any other questions, please let us know.
>
> Sincerely,
>
> Authors

---

> ### Author Response · Authors · 2024-11-20
>
> >Q2
>
> The result is a property of any MDP, for both single agent and multi agents. When we talk about the optimal policy in MARL, we mean the optimal joint policy. There exists at least one deterministic optimal joint policy in a multi-agent MDP. Your understanding of joint policy is correct, which means the global action $<a_1,a_2,....,a_N>$.
>
> > Q3
>
> You might have confused theory with practice. In Lemma 4, we talk about theory. In theoretical proof, we should go through all kernels to guarantee the convergence and optimality, just like that Q-learning assumes going through all state-action pairs. Since the analysis of Q-learning **requires countable state-action pairs**, the kernels will not be infinite in theoretical proof with deterministic policy. In the case of infinite many kernels exist, it is beyond the scope of theoretical proof, and we only talk about practice. The experimental results can evaluate the performance of BQL in that case.

---

> ### Comment · Reviewer_bFNq · 2024-11-20
>
> My major concern is the lack of quantitative results, which raises questions about the impact and significance of the theoretical findings.
>
> (1) The convergence presented in the paper is asymptotic, based on the fact that improvement only occurs when the best kernel is selected. However, how likely is this to happen? When the state/action space is large, the probability of selecting the best kernel becomes smaller, potentially leading to significantly slower convergence rates. Sure the algorithm will converge, but is it linearly fast? Or could the rate become exponentially slow as $S,A$ scale? The lack of a quantitative characterization makes the impact of this result unclear. The asymptotic convergence is expected and such a result alone is not informative.
>
> (2) What is the total computational complexity of the simplified operator until it converge to the optimal policy? Intuitively, the average number of ineffective updates (i.e., when the best kernel is not selected) appears to match the total number of kernels. If this is the case, the computational cost is not reduced compared to the best possible operator (in expectation sense), making the claim of reduced cost unconvincing. A quantitative characterization would make these discussions much clearer.
>
> For the current version, the claims are not fully supported by the theoretical results. Furthermore, the proof offers limited insights or novelty, and it provides little practical guidance (e.g., expected computational costs of the algorithm implementations). Therefore, I feel the paper is not yet ready for publication.

---

> ### Author Response · Authors · 2024-11-21
>
> We agree the complexity of BQL is high, but that is because the complexity of fully decentralized MARL is high (https://arxiv.org/pdf/1301.3836). To the best of our knowledge, no other algorithm shows lower complexity or faster convergence than BQL in this problem.
>
> Why do you think the proof offers limited insights or novelty? The proof shows that BQL is the first algorithm to guarantee convergence and optimality in fully decentralized MARL. Is it not novel enough? Why do you think the proof provides little practical guidance? The derived algorithm from the proof achieves significant performance gain in experiments.
>
> Do you really think the complexity or convergence rate provide any practical guidance? Think about the successful RL applications, AlphaGo and ChatGPT, which one is related to the analysis of complexity or convergence rate? In large-language model, the space of state-action pair is incredibly large, the complexity analysis is ridiculous in front of it. Excessive focus on complexity analysis is a significant obstacle to the practical application of RL, and a waste of life.

---

> > ### Author Response · Authors · 2024-11-24
> >
> > Dear reviewer bFNq,
> >
> >
> > Are you satisfied with our response? As the rebuttal deadline is approaching, it is urgent to figure out whether the complexity analysis benefits the application of RL or is merely a universal reason for rejection?

---

> > > ### Comment · Reviewer_bFNq · 2024-11-24
> > >
> > > I believe this is due to differences in my research philosophy and that of the authors. I do not intend to comment further on this. However, regarding the paper itself, my main concern is the lack of quantified results, which are essential to substantiate the authors' claims.
> > >
> > > 1. Complexity Analysis: The stochastic algorithm is claimed to have lower complexity than the best operator. However, the paper lacks a formal complexity analysis to support this claim. The analysis in Appendix B is neither rigorous nor convincing. It is possible that the total complexity is not actually improved compared to the best possible algorithm. The authors claim the high complexity is due to MARL itself, but still unclear with actual equations.
> > >
> > > 2. Scalability: The scalability of the algorithm is unclear. Without a thorough complexity analysis, it is difficult to determine whether the algorithm can handle large-scale problems or if it is limited to small-scale ones. For instance, consider a single-agent RL problem: while comparing the performance of every deterministic policy one by one may work well for small-scale problems, it becomes prohibitively expensive for large-scale problems. Although the authors claim that the experiments indicate scalability, the results are limited to specific environments and do not conclusively demonstrate representativeness. If no quantified results are available, more extensive experimental results may be needed.
> > >
> > > 3. Focus on Easier Setting: The paper is primarily focused on a simplified setting (only one optimal policy). The studies presented in Appendix D are neither comprehensive nor representative.
> > >
> > > Due to these weaknesses in the results and analysis, I am inclined to reject this paper.

---

> ### Author Response · Authors · 2024-11-25
>
> Please check our response to Reviewer sXth. The stochastic algorithm is proposed for neural network implementation not for complexity. The best operator cannot be written as a loss function for a neural network because each agent has to compute the expected values of all possible transition probabilities, so the stochastic algorithm is needed.
>
> It seems that you deeply believe the complexity is necessary to RL application, so do you know the complexity of PPO in general cases?
>
> You said BQL is poor in complexity analysis, scalability, experiments, and easier setting. However, the previous methods in decentralized MARL even cannot guarantee convergence and optimality, so the complexity cannot be discussed. And the previous methods show worse scalability, easier settings (see related work), and more limited experimental results than BQL. After you read these papers, do you think the previous methods should also be rejected or this research field should be vanished? Frankly, this research field has already been vanished.

---

> ### Author Response · Authors · 2024-11-25
>
> Dear reviewer bFNq,
>
>
> We feel that you are requiring us to **thoroughly solve** the problem of decentralized MARL. But the aim of this paper is to obtain a practical algorithm from the analysis of convergence and optimality. Complexity analysis usually stands alone as a separate paper. And we have to argue that complexity analysis is not a reason for rejecting an ICLR paper, because ICLR is not a conference for mathematics. You mentioned that we focus on a simplified setting with only one optimal policy, however, whether the optimality can be guaranteed with multiple optimal policies in decentralized MARL is still an open problem. And the study on this problem called coordination also usually stands alone as a separate paper. You mentioned the experiments are limited, but you also said that you do appreciate the experiment part in the main review. You mentioned that the problem of MARL is definitely important in the main review, but have you ever thought about the reason why the previous methods on this important problem are so limited? We believe that when we evaluating a research paper, we should compare it with the previous methods, instead of requiring it to solve all aspects in a research field.

---

> > ### Comment · Reviewer_bFNq · 2024-12-02
> >
> > I appreciate the authors' response. I understand that it is impractical to solve all problems in one paper. I will keep this in mind during the discussion period.

---

### Official Review · Reviewer_HS25 · 2024-10-18

**Soundness:** 1
**Presentation:** 2
**Contribution:** 2
**Rating:** 1
**Confidence:** 4

**Summary:**

This paper presents a new take on decentralized Q-learning, in which each agent (independently) updates its value function by pretending that the other agents act optimally with respect to its current estimate of the value function. A simplification of this procedure is proposed which the authors show also leads to desirable convergence properties. Experiments contrast the proposed method with relevant baselines.

**Strengths:**

* The main idea of having each agent “imagine” a best-case scenario of what the other agents do is a very clean way to synchronize the independent Q-learning approach which could otherwise fail to converge.
* For the most part, the proofs are logical and easy to follow. (See question below.)

**Weaknesses:**

* Some syntax errors on line 70: no whitespace after “MDP” and also the <> symbols should probably be () for a tuple as is standard in the MDP literature.
* Many instances of grammatical issues, e.g. missing “the” in a sentence.
* The simplified operator seems like it will be egregiously inefficient, particularly in larger state/action spaces. Isn’t it effectively just random search?
* The sentence “but the converged equilibrium may not be the optimal one when there are multiple Nash equilibria” on line 322 only makes sense in (a) games where agents share a common objective — otherwise what is the measure of optimal — and (b) is not really a critique for non-convex games where methods would only at best be able to find local equilibria and there is no good way to find the “best” such equilibrium. Similar comment for the statement about Hysteretic IQL on line 325.
* “slow learning rate to the value punishment” on line 324 must be a typo of some kind
* “using mean and standard” on line 345 is a typo
* “action space of each agent is 4” -> another typo on line 351
* More details should be provided about the distribution from which transition kernels and reward functions are sampled (cf line 354). Depending on the variance of these distributions, using only 20 samples could easily give a very poor statistical estimate of performance. Since the standard deviations look fairly small (at least in Figure 1), I am guessing that these distributions are fairly well structured. It would be useful to understand that structure so as to better appreciate the testing scenario and results.
* “std” on line 356 should be spelled out
* I do not follow all the issues that are being discussed in the paragraph ending on line 374. This discussion should be substantially rewritten for clarity.
* “wildly adopted” -> typo on line 301. I presume it should be “widely” but if that is the case, why is there only one reference here?
* “MPE” is not defined anywhere as far as I can remember. Cf. line 341 and section 4.2. From the description in section 4.2, this is not actually a differential game since time is not a continuous variable. Cf. the standard texts by Rufus Isaacs and Tamer Basar & Geert Olsder. The proper term would be “dynamic game,” which includes discrete-time problems.
* “In continuous environments, BQL and baselines are built on DDPG.” (Line 428) -> this should be explained far more carefully so that readers can appreciate how this choice influences the results
* In Figure 2, the results are not easily interpretable due to the color choices being so similar for the confidence intervals, and the intervals themselves being so wide for some methods.
* All of the subfigure captions in Figures 3-5 are confusing. Without clear explanations of every single experiment, results are impossible to interpret and certainly cannot be trusted by a serious reader.
    * Some of the descriptions of these subfigures in section 4.4 even make it sound like these are tasks where different agents have different objectives; clearly that does not conform to the present paper setting, so something is additionally confusing here.

**Questions:**

* Surely the claim of global optimality in line 62 is restricted to tabular cases, right? Such broad claims should be avoided (unless somehow they are true) in order to ensure the work is not misunderstood by a casual reader. Please clarify the setting in which this claim is valid.
* Why can’t we treat all agents as one “super agent” whose action space is the product space of the individual agents, and conclude that because (tabular) Q learning would converge for the “super agent,” it will also for the individual Q learners? If we did value iteration instead of Q learning would it work? An explanation here would probably not depend upon (non)stationarity arguments, I think. At the very least, the authors should provide a reference for the statement of non-convergence on line 87.
* I do not follow the step from line 142 to line 144. Is it generally true that max_x f(x) - max_x g(x) \ge f(x’) - g(x’) for a generic x’ (here, x’ is analogous to a_i^{‘*})? I can certainly find counterexamples in general, so what is special about this case which admits this inequality? Please explain what is going on here more carefully.
* I am not clear on why we need to “explore” all state/action pairs with a deterministic policy (looking at line 236). Can’t we just enumerate them in a tabular problem? Please explain why the exploration is needed and enumeration does not suffice.
    * This relates to the italicized sentence at the end of section 2.

---

> ### Author Response · Authors · 2024-11-14
>
> > Surely the claim of global optimality in line 62 is restricted to tabular cases, right? Please clarify the setting in which this claim is valid.
>
> In the Q-learning field, all discussions of convergence and optimality of are based on tabular cases since the convergence of Q-learning cannot be guaranteed in neural network approximation, as shown in (https://arxiv.org/pdf/1903.08894).
>
> > Why can’t we treat all agents as one “super agent” whose action space is the product space of the individual agents? At the very least, the authors should provide a reference for the statement of non-convergence on line 87
>
> In fully decentralized  MARL, each agent does not know other agents' action space, so the action space cannot be the product space of the individual agents. The non-convergence of IQL (line 87) is common knowledge and is discussed in all related work listed.
>
> > I do not follow the step from line 142 to line 144. Is it generally true that max_x f(x) - max_x g(x) \ge f(x’) - g(x’) for a generic x’ (here, x’ is analogous to a_i^{‘*})? I can certainly find counterexamples in general, so what is special about this case which admits this inequality?
>
> $\max_x f(x) - \max_x g(x) \ge f(x’) - g(x’)$ where $g(x’) = \max_x g(x)$. Since $\max_x f(x)$ is the max of $f(x)$, $\max_x f(x) \ge any f(x’)$. How to build the counterexamples?
>
> > I am not clear on why we need to “explore” all state/action pairs with a deterministic policy (looking at line 236). Can’t we just enumerate them in a tabular problem? Please explain why the exploration is needed and enumeration does not suffice.
>
> Of course, we can enumerate them. Enumeration is a special case of exploration. Our algorithm shows how to explore them if enumeration is not allowed.

---

> > ### Author Response · Authors · 2024-11-24
> >
> > Dear reviewer HS25,
> >
> >
> > Waiting for your counterexamples.

---

> > > ### Author Response · Authors · 2024-11-26
> > >
> > > Dear reviewer HS25,
> > >
> > > Since the discussion period has been extended, we have enough time to correct your bias towards our paper. Can we begin with the questions you've raised?

---

> > > > ### Author Response · Authors · 2024-11-27
> > > >
> > > > Dear reviewer HS25,
> > > >
> > > > Based on your review, we guess that you are not familiar with RL. For example, even though we have already provided the reference, you still do not know that MPE is one of the most popular testbeds in MARL. You do not know that DDPG is the implementation of Q-learning in continuous environments. The questions you've raised show that you are unfamiliar with the proof of the contraction in Q-learning. The examples listed above are basic knowledge in RL, and such high confidence in such easy questions seems a bit funny.

---

> > > > > ### Author Response · Authors · 2024-11-27
> > > > >
> > > > > Dear reviewer HS25,
> > > > >
> > > > > If it's inconvenient for you to continue discussing the questions you've raised, let's change the topic. What do you think is the fatal flaw of BQL?

---

> ### Author Response · Authors · 2024-11-30
>
> Dear reviewer HS25,
>
> Since we haven't received your response, can we assume that you think **there is no fatal flaw in BQL**?

---

### Official Review · Reviewer_sXth · 2024-10-30

**Soundness:** 3
**Presentation:** 3
**Contribution:** 3
**Rating:** 6
**Confidence:** 3

**Summary:**

This paper studies decentralized multi-agent Q-learning where every agents share the same state and observes only local action. Resolving non-stationary due to the joint-policy, it is difficult to establish convergence of to a joint optimal policy is an important problem in MARL. The authors propose a best-possible operator, which basically solves the joint opitmal Bellman equation for $i$-th agent:
$$   Q^i(s,a_i)=\max_{a_{-i}}\mathbb{E}_{s^{\prime}\sim P(\cdot\mid s,a_i,a_{-i} )}[r+\gamma Q^i(s,a_i)].$$
The authors solve the above equation in decentralized manner, and its extension to practical algorithm is studied.

**Strengths:**

1. The authors proposed a new decentralized algorithm that could give new insights into the community. Even though the proposed algorithm has limitations, finding a global joint optimal policy in a decentralized manner seems to be a contribution.

2. The overall experimental result seems positive. It shows better or comparable results to existing ones including MA2QL, I2Q, H-IQL, IQL.

**Weaknesses:**

1. The memory space to store $P(\cdot\mid s,a_i,a_{-i})$ requires at least $O(|\mathcal{A}_{-i}|)$ space, which scales exponentially at the order of each action space. This makes the algorithm difficult to scale as number of agents increase. If we use function approximation, or somewhat similar methods to reduce this problem, will the arguments of this paper be still valid?

2. The search over all possible $\pi_{-i}(a_{-i},s)$ for evert state $s$ and $i$ seems to be quite a burden. It at least requires $N \prod_i |\mathcal{A}_i|$, which scale exponentially due to the joint action space. Even though the authors proposed a version in (7) to reduce the search time, I do not think this gives a meaningful cut in the search time in theoretical sense. Can the authors provide theoretical advantage in terms of search time for the proposed method?

3. A closely related work [1] deals with theoretical convergence of independent Q-learning. Please provide comparison with the work in sense that how the setting is different, and pros and cons.


4. The algorithm does not seem to be scalable.

**Questions:**

1. In Lemma 1, why do we need the condition of existence of only one optimal policy?

2. There is typo in the caption of Figure 3 : "Mojoco"->"Mujoco".

3.  In Section 2.4, $S^m_i$ has not been explained previously.

---

> ### Author Response · Authors · 2024-11-14
>
> >Scalability
>
> First, we have to argue that the complexity converging the optimum in fully decentralized MARL is high. The complexity exponentially grows as the number of agents increases, **so any algorithm for fully decentralized MARL must have exponential complexity**.  Second, in neural network implementation, we only maintain one replay buffer for each agent, without explicitly storing $P(\cdot|s, a_i a_{-i})$ or searching over all possible $\pi_{-i}$. So the sample collection of BQL is consistent with other baselines. Third, in Figure 3(d), BQL can achieve significant performance improvement when the agent number is large (17 agents), which shows the scalability of BQL is better than other baselines. Last, the aim of (7) is not to reduce the search time, but for the convenience of neural network implementation. The operator (6) has the same complexity as the simplified version (7,8), but the operator (6) cannot be written as a loss function for a neural network because each agent has to compute the expected values of all possible transition probabilities.
>
> > A closely related work [1] deals with theoretical convergence of independent Q-learning. Please provide comparison with the work in sense that how the setting is different, and pros and cons.
>
> Sorry, but the citation [1] seems to be missed.
>
> > In Lemma 1, why do we need the condition of existence of only one optimal policy?
>
> Without this condition, BQL can still converge to the optimal value but does not know how to select the optimal joint action. Look at the matrix game in Figure 11, there are two optimal policies (1,2) and (2,1). For each agent, actions 1 and 2 have the same highest optimal learned value. If each agent arbitrarily selects one of the optimal independent actions, the selected joint action might not be optimal (policy (1,1)). We have discussed the problem of multiple optimal policies in Appendix D.
>
> > In Section 2.4, $S^m_i$ has not been explained previously.
>
> We have claimed that $S^m_i$ is a subset of states randomly selected by agent $i$ at the epoch $m$ at line 238.

---

> > ### Comment · Reviewer_sXth · 2024-11-15
> >
> > Thank you for the detailed response. The missing citation is the following:
> >
> > [1] Jin, Ruiyang, et al. "Approximate Global Convergence of Independent Learning in Multi-Agent Systems." arXiv preprint arXiv:2405.19811 (2024).

---

> > > ### Author Response · Authors · 2024-11-15
> > >
> > > Thanks for pointing out this related work. We will add it into the revision. There are two significant difference between our work and the citation:
> > >
> > > 1. We propose a novel algorithm BQL for the convergence and optimality in decentralized MARL, while the citation only provides the analysis of global optimality gap in IQL.
> > >
> > > 2. The citation proves that IQL cannot guarantee the optimality, where the optimality gap is bounded by dependence level, which describes the influence between agents. It is straightforward to understand that the large dependence level  leads to large optimality gap. However, BQL can guarantee the optimality regardless of the dependence level.

---

### Official Review · Reviewer_3pSK · 2024-11-06

**Soundness:** 3
**Presentation:** 3
**Contribution:** 3
**Rating:** 6
**Confidence:** 3

**Summary:**

This paper considers decentralized training decentralized play for multi-agent reinforcement learning. A fully decentralized learning is proposed based on a best possible operator proposed in this paper. Though this operator is computationally expensive, it leads to convergence despite the nonstationarity caused by other players. A simplified operator is also proposed and the convergence and optimality based on this simplified operator is also provided. Simulation results demonstrate the performance of the proposed algorithm.

**Strengths:**

1. The paper is quite novel and original since it proposed a novel operator to enable convergence despite the nonstationary environment caused by other players in a decentralized multi-agent and stochastic setting.
2. The paper is very well written. The algorithm is explained clearly and the simulation results are easy to follow.
3. The results are significant since it addressed a long lasting open question on MARL.

**Weaknesses:**

1. It can be restrictive to assume there is only a unique optimal policy. How does the proposed algorithm perform when there are multiple optimal policies?

2. Can the authors provide explicit theorems and proofs for the convergence and optimality of BQL  for both the tabular case and the neural network case?

**Questions:**

See above.

---

> ### Author Response · Authors · 2024-11-14
>
> Thank you for reviewing our paper. We will carefully address your questions.
>
> > It can be restrictive to assume there is only a unique optimal policy. How does the proposed algorithm perform when there are multiple optimal policies?
>
> In Appendix D, we have discussed how to deal with the case with multiple optimal policies and provided a one-stage matrix game in Figure 11. BQL can select coordinated actions, though the value gap between the optimal policy and suboptimal policy is so small.
>
> > Can the authors provide explicit theorems and proofs for the convergence and optimality of BQL for both the tabular case and the neural network case?
>
> We have proven the convergence and optimality of tabular case. The convergence of Q-learning cannot be guaranteed in function approximation methods, e.g., DQN with a neural network, as shown in (https://arxiv.org/pdf/1903.08894).

---

### Author Response · Authors · 2024-12-02

Dear ACs, SACs, and PCs,


It seems that our paper will be rejected, however, we have a question that we hope can be addressed in the meta review. Decentralized MARL is an important problem. No reviewer questions that BQL is the first algorithm to guarantee convergence and optimality in decentralized MARL with both deterministic and stochastic environments, and all reviewers seem to appreciate the experiment part. The first algorithm shows the novelty and contribution. The proof and the experiments verify the correction. So, what is the fundamental reason why this paper cannot meet the bar for ICLR, especially when baselines that are theoretically and experimentally inferior to BQL have already been accepted by top conferences, and the negative feedback raised by the reviewers also applies to those baselines?


Sincerely,

Authors

---

### Meta-Review · Area_Chair_Ghnu · 2024-12-10

**Metareview:**

This paper studies decentralized reinforcement learning over networked agents, based on a best-possible operator. The studied problem is important and the algorithm design contains new ideas. In particular, the use of the best-possible operator in this decentralized multi-agent RL setting is interesting, the paper is overall easy to follow, and the experiments are compelling in general. However, there were some concerns regarding the rigor and clarity of the technical contributions, as well as the presentation/writing of the paper. Theoretical contributions have taken up quite some space in the paper, but seem to be not significant and strong enough to clear the bar for acceptance, especially given some strong assumptions required in the theorems, which were also stated relatively loosely/informally. In fact, for the core algorithm BQL, there were no theoretical guarantees (as per Sec. 2.4). I suggest the authors incorporate the feedback in preparing the next version of the paper.

**Additional Comments On Reviewer Discussion:**

There were multiple rounds of discussions between the reviewers and the authors. The key concerns were regarding the significance and novelty of the technical results, especially under strong assumptions, as well as the rigor and clarity of the exposition. The authors have responded, but did not fully address all the comments, with some behaviors not very appropriate in my book. As the paper did sell "theoretical contributions" as one big part of their results, it would have been stronger to make the theoretical results more solid and compelling.

---

### Decision · Program_Chairs · 2025-01-22

Reject